# Acanthosis nigricans as a composite marker of cardiometabolic risk and its complex association with obesity and insulin resistance in Mexican American children

**Juan C. Lopez-Alvarenga**[1☯]*, **Geetha Chittoor**[2☯], **Solomon F. D. Paul**[3☯], **Sobha Puppala**[4], **Vidya S. Farook**[1], **Sharon P. Fowler**[5], **Roy G. Resendez**[1], **Joselin Hernandez-Ruiz**[6], **Alvaro Diaz-Badillo**[1], **David Salazar**[7], **Doreen D. Garza**[7], **Donna M. Lehman**[5], **Srinivas Mummidi**[1], **Rector Arya**[1], **Christopher P. Jenkinson**[1], **Jane L. Lynch**[8], **Ralph A. DeFronzo**[5], **John Blangero**[1], **Daniel E. Hale**[9], **Ravindranath Duggirala**[1]

1 Department of Human Genetics and South Texas Diabetes and Obesity Institute, University of Texas Rio Grande Valley, Edinburg and Brownsville, TX, United States of America, 2 Biomedical and Translational Informatics, Geisinger Health System, Danville, PA, United States of America, 3 Department of Human Genetics, Sri Ramachandra Institute of Higher Education and Research, Porur, Chennai, Tamil Nadu, India, 4 Department of Internal Medicine, Wake Forest University, Winston-Salem, NC, United States of America, 5 Department of Medicine, University of Texas Health San Antonio, San Antonio, TX, United States of America, 6 Department of Pharmacology, Hospital General de Mexico "Dr. Eduardo Liceaga", Mexico City, Mexico, 7 Border Health Office, College of Health Professions, University of Texas Rio Grande Valley, Edinburg, TX, United States of America, 8 Department of Pediatrics, University of Texas Health San Antonio, San Antonio, TX, United States of America, 9 Pediatric Endocrinology and Diabetes, Penn State University, Hershey, PA, United States of America

☯ These authors contributed equally to this work.
* juan.lopezalvarenga@utrgv.edu

**Data Availability Statement:** We provided supplementary files relating to the minimal data

## Abstract

### Aim

Acanthosis nigricans (AN) is a strong correlate of obesity and is considered a marker of insulin resistance (IR). AN is associated with various other cardiometabolic risk factors (CMRFs). However, the direct causal relationship of IR with AN in obesity has been debated. Therefore, we aimed to examine the complex causal relationships among the troika of AN, obesity, and IR in Mexican Americans (MAs).

### Methods

We used data from 670 non-diabetic MA children, aged 6–17 years (49% girls). AN (prevalence 33%) severity scores (range 0–5) were used as a quasi-quantitative trait (AN-q) for analysis. We used the program SOLAR for determining phenotypic, genetic, and environmental correlations between AN-q and CMRFs (e.g., BMI, HOMA-IR, lipids, blood pressure, hs-C-reactive protein (CRP), and Harvard physical fitness score (PFS)). The genetic and environmental correlations were subsequently used in mediation analysis (AMOS program). Model comparisons were made using goodness-of-fit indexes.

sets corresponding to the results including the matrices of genetic and environmental correlations and the data of residuals (i.e., residuals obtained for a given phenotype after adjusting for covariate effects) that we used for the analyses. None of them have any identification codes.

**Funding:** This study was supported RD received grants: R01 HD049051/HD049051-5S1 [ARRA], HD041111, DK053889, DK042273, DK047482, MH059490, P01 HL045522, K01 DK064867, M01-RR- 01346, and Veterans Administration Epidemiologic Grant. The funders had no role in study design, data collection and analysis, decision to publish, or preparation of the manuscript. Some of the investigators received salaries from some of the grants as follows: R01 HD049051/HD049051-5S1 [ARRA] (RD, DEH, JB, RAD, CPJ, RA, SP, VSF, SPF, RGR), HD041111 (RD, JB, RAD, CPJ, RA, RGR), DK053889 (RD, JB, CPJ, SP, VSF, SPF, RGR), DK042273 (DML, RD, JB), DK047482 (DML), MH059490 (JB), P01 HL045522 (JB, RD), and Veterans Administration Epidemiologic Grant (RAD, CPJ). The AT&T Genomics Computing Center supercomputing facilities used for this work were supported in part by a gift from the AT&T Foundation and with support from the National Center for Research Resources Grant Number S10 RR029392. This investigation was conducted in facilities constructed with support from Research Facilities Improvement Program grants C06 RR013556 and C06 RR017515 from the National Center for Research Resources of the National Institutes of Health.

**Competing interests:** The authors have declared that no competing interests exist.

## Results

Heritability of AN-q was 0.75 (p<0.0001). It was positively/significantly (p<0.05) correlated with traits such as BMI, HOMA-IR, and CRP, and negatively with HDL-C and PFS. Of the models tested, indirect mediation analysis of BMI→HOMA-IR→AN-q yielded lower goodness-of-fit than a partial mediation model where BMI explained the relationship with both HOMA-IR and AN-q simultaneously. Using complex models, BMI was associated with AN-q and IR mediating most of the CMRFs; but no relationship between IR and AN-q.

## Conclusion

Our study suggests that obesity explains the association of IR with AN, but no causal relationship between IR and AN in Mexican American children.

## 1. Introduction

Acanthosis nigricans (AN) is an epidermal disorder of velvety thickening that primarily affects the axillae, posterior neck fold, flexor skin surfaces, and umbilicus [1,2]. AN has been associated with obesity and insulin resistance (IR) [1,2]; and, has been used as a marker for assessing type 2 diabetes (T2DM) risk in children [3]. For example, individuals with body mass index (BMI) greater than the 98th percentile were found to have a 62% prevalence of AN [4]; and, the presence of AN in Mexican individuals could predict IR with 66.7% sensitivity, which was shown to be influenced by the skin phototype [5]. Thus, AN has been used as a screening tool for identifying children at risk of insulin resistance (IR), T2DM, and cardiometabolic risk factors (CMRF) [2,6–8], in order to prevent progression of these disease conditions into adulthood.

There are remarkable ethnic disparities in the prevalence of AN. For example, in the United States (US), according to a cross-sectional study of young persons (the Research in Outpatient Settings Network), the prevalence of AN was found to be higher in Hispanics (19%) and American Indians (28%) compared to non-Hispanic whites (3%) [9]. As we reported previously, the occurrence of AN in Mexican Americans (MA) children and adolescents from families at high risk for obesity, CMRFs, and T2DM was 33% [10]. The diversity in clinical manifestations of AN can be attributed to the influences of genetic and environmental factors and their interactive influences on AN. Aside from these observations, it is not known whether AN and its clinical correlates, particularly obesity and IR, are causally related. Recently, Hudson et al. [11], using data from a community sample of adolescents with obesity, found that AN (or severity) was a poor marker of IR or cardiometabolic risk after adjusting for obesity (i.e., BMI).

Therefore, since the direct causal relationship of IR with AN in obesity is debatable, we aimed to assess genetic and environmental correlations between AN and CMRFs and examine the complex causal relationships among the troika of AN, obesity, and IR using data from children and adolescents from MA families. In contrast to population studies comprising of unrelated individuals, family-based studies such as ours provide opportunities not only to determine genetic basis of given traits, but also to assess phenotypic, genetic, and environmental correlations among correlated traits such as IR, AN, and obesity. Here, we analyzed the association of AN severity score with IR, obesity measures and other CMRFs. To assess the effects of IR and obesity on AN and other CMRFs, a mediation model was employed using information from the matrices of genetic and environmental correlations.

## 2. Materials and methods

This part of the study was made with unidentified data bases. Briefly, all research procedures were approved by the Institutional Review Board of the University of Texas Health Science Center at San Antonio. Written informed consent was obtained from one or both parents of each child. The San Antonio Family Assessment of Metabolic Risk Indicators in Youth (SAFARI) study, a community-based family study, was designed to identify signs of metabolic syndrome (MS) and future disease risk in MA children and adolescents in San Antonio, TX and surrounding areas, and to examine their genetic basis [10]. As part of the SAFARI study, 673 children and youth, aged 6–17 years old, were recruited from large, predominantly lower-income MA families at increased risk of obesity and T2DM. In specific, the SAFARI participants are the offspring of adults participated in three of our past or ongoing MA family studies: San Antonio Family Diabetes/Gallbladder Study (SAFDGS, n = 126), San Antonio Family Heart Study (SAFHS, n = 373) and, Veterans Administration Genetic Epidemiology Study (VAGES, n = 174), which were initiated in the early 1990s. However, based on a subset of the SAFARI children (~70%) for whom the information on birth places of mothers (and maternal grandparents) and fathers (and paternal grandparents) was available, some parents (i.e., mother or father) of the SAFARI children were born outside of the US, almost exclusively from Mexico as follows: Mothers = ~8% (grandmother = 19% and grandfather = 22%) and Fathers = ~12% (grandmother = 23% and grandfather = 26%). As reported previously [10], briefly, these 673 children generated 3,664 relative pairs including various types of relatives. Some examples are given as follows: Siblings = 383, 1st cousins = 550, 1st cousins once removed = 234, 2nd cousins = 661, 2nd cousins once removed = 512, 3rd cousins = 662, and 3rd cousins once removed = 137. Three of 673 children were found to have T2DM, based on our clinic examinations; these children were excluded from the analyses. Additional details on study and recruitment procedures were previously reported and discussed in Fowler et al. [10] This study considered data from a total of 670 non-diabetic SAFARI study participants (S1 Table).

### 2.1. Phenotypic and covariate data

An extensive battery of clinical tests and interviews was administered to SAFARI participants to collect information on medical history, family history, demographic, phenotypic, and environmental variables related to MS/CMRFs using standard protocols, as reported previously by Fowler et al. [10] The CMRFs considered for the current study included the following: body mass index [BMI: kg/m$^2$], waist circumference (WC), fasting glucose (FG), fasting insulin (FI), homeostasis model assessment of insulin resistance (HOMA-IR; using information from FG and FI), high density lipoprotein cholesterol (HDL-C), triglycerides (TG), systolic (SBP) and diastolic (DBP) blood pressure, high-sensitivity C-reactive protein (CRP), and Harvard physical fitness score (PFS) [10].

In addition, the following phenotypic information described by Fowler et al. [10] was used for this study. BMI percentiles by sex and age were obtained from NHANES III to define obesity categories: normal = below the 85th percentile; overweight = ≥85th percentile and <95th percentile; and, obese = ≥95th percentile. Prediabetes was defined using information from impaired fasting glucose (IFG: Fasting plasma glucose (FPG) ≥ 100 and <126 mg/dl), impaired glucose tolerance (IGT: 2-h PG ≥ 140 and <200 mg/dl) or both, following the ADA guidelines [12]. Following our previous report, MS was defined as presence of ≥3 of the following 6 dichotomized MS component traits [10]: 1. Abdominal obesity: waist circumference ≥90th percentile for age, sex and MA ethnicity; 2. Hypertriglyceridemia: triglycerides ≥110 mg/dl and 3. Low HDL-C: ≤40 mg/dl; 4. Elevated blood pressure (SBP and/or DBP): ≥90th

percentile for height, age and sex; 5. Glucose intolerance: IFG, IGT, or both as defined above; and 6. Hyperinsulinemia: >75th percentile for total SAFARI cohort (≥16.25 µIUml). Using a scale previously developed and validated by our group, presence/severity of AN on the neck was evaluated [13]. 16 AN severity score ranged from 0 to 5, and the dichotomous trait AN was defined as AN severity score ≥ 2. AN severity score was used as a quasi-quantitative trait for analysis and is referred to as AN-q. Pubertal status was assessed using Tanner staging.

## 2.2. Variance components analysis

The univariate and bivariate genetic analysis were conducted using the variance components approach (VCA) as implemented in the program SOLAR [14]. The heritability of AN-q was determined using the VCA. In a simple model, variances or covariances between relatives as a function of the genetic relationships can be specified, and the proportion of phenotypic variance that is attributed to (additive) genetic effects (i.e., heritability: $h^2$) can be estimated from the components of variance [14,15]. 17 For such a model, the covariance matrix for a family ($\Omega$) is given by: $\Omega = 2\Phi\sigma^2_g + I\sigma^2_e$, where $\Phi$ is the kinship matrix, $\sigma^2_g$ is the genetic variance due to additive genetic effects, I is the identity matrix, and $\sigma^2_e$ is the variance due to individual-specific environmental effects. A likelihood ratio test was used to test whether the heritability of AN-q was significant. Adjustment for covariate (e.g., age, sex, and pubertal status) influences were made in the analysis. Given the sample size (e.g., AN-q) and pedigree structure used for this study, we will have 80% power to detect heritabilities as small as 0.25. The phenotypic, genetic, and environmental correlations between two traits (e.g., AN-q and BMI) were determined using bivariate genetic analysis. This is an extension of the VCA where the phenotypic correlation ($\rho_P$) between a pair of quantitative traits (e.g., AN-q and BMI) can be partitioned into additive genetic ($\rho_G$) and environmental ($\rho_E$) components [15,16]. A likelihood ratio test was used to test whether a given correlation was significant. The additive genetic correlation ($\rho_G$) is a measure of the shared genetic basis of the two traits (i.e., pleiotropy), while the random environmental correlation ($\rho_E$) is a measure of the strength of the correlated response of a trait-pair to non-genetic factors. All analyses included adjustments for covariate (e.g., age, sex, and pubertal status) influences. Regarding power to detect genetic correlations in our data (e.g., AN-q and DBP trait pair), we will have 86% power for detecting a genetic correlation as small as 0.30. If non-normality of a given trait was an issue, we used appropriate transformations (i.e., log transformation or inverse normal transformation).

## 2.3. Mediation analysis

To assess causal relationships between traits, we used the mediation analysis [17]. In a simple model, a mediator (M) is a variable, which is in a causal sequence between an independent variable X and a dependent variable Y (i.e., X→M→Y). To begin with, for example, the structure considered BMI has effect on AN and IR, in turn IR affects AN and CMRFs. Using the genetic and environmental correlations (obtained after adjusting covariate effects such as age, sex and pubertal status) determined by bivariate genetic analysis, standardized beta coefficients were calculated to compare effects of loadings on clinical traits considered for this study. Models of full mediation, indirect mediation, and partial mediation were used for the given sets of data and tested using the goodness-of-fit statistics Akaike information criterion (AIC), and Bayes information criterion (BIC). The model selection between the competing models was made based on AIC and BIC criteria by making balance between goodness of fit and parsimony. The models with lower AIC and BIC were considered as the best models. All statistical analyses were performed using IBM® SPSS AMOS load 23. To assess power, the statistical analysis considered the effect of an independent variable (X) on a dependent variable (Y) through a

mediating variable (M). The path diagram considered small effects for three parameters: α is the estimate of the effect of X on M; β is the estimate of the effect of M on Y adjusted for X, and the $\tau^*$ parameter corresponds to the total effect of X on Y adjusted for M (and compared with τ without M adjustment). The smallest effect for a difference ($\tau - \tau^*$) that we considered was 0.14 for each parameter; thus, the sample needed for 80% statistical power was 550 subjects [18].

## 3. Results

As stated above, of the 673 children, 3 of them with T2DM were excluded and only data from 670 nondiabetic children were used for the analyses. The 673 children represented 401 nuclear families (~2 children per nuclear family; range = 1–5 children) and generated 3,664 relative pairs [10]. The characteristics of the 670 nondiabetic children were reported previously by Fowler et al. [10] However, description of age, sex, pubertal status, overweight, obesity, prediabetes, MS, AN and AN-q measures pertaining to this study as outlined in Table 1. We found disturbingly high rates of overweight (19.1%), obesity (34%), pre-diabetes (13%), MS (19%), and AN (33%). The calculated heritability of AN severity score was $0.75 \pm 0.11$ ($3.6 \times 10^{-12}$).

The occurrence of AN increased by obesity status: normal weight (11%), overweight (53%), and obese (65%); by prediabetes status (47%); and by MS status (67%), respectively (Table 1). For example, based on odds ratios (ORs), children with AN are approximately 9/10 times more likely to be obese/overweight, 6 times more likely to have MS, and 2 times more likely to have prediabetes. Likewise, more or less similar patterns of relationships between AN-q and selected quantitative traits BMI, HOMA-IR, and CRP were observed (Fig 1). As expected, PFS was inversely associated with AN-q. As can be seen, AN-q exhibited linear trend with BMI, HOMA-IR, CRP and PFS (ANOVA for linear trend $p < 0.001$. Fig 1). As shown in Table 1, heritability ($h^2$) estimate for AN-q was found to be significantly ($P < 0.0001$) and highly heritable ($h^2$: 0.75). The results of phenotypic ($\rho_P$), genetic ($\rho_G$), and environmental ($\rho_E$), correlations between AN-q and obesity/CMRFs are given in Table 2. As can be seen, with an exception of FG, all of the phenotypic correlations were significant ($\rho_P$ range: -0.35 [AN-q and PFS] to 0.66 [AN-q and WC]). Interestingly, excluding FG, the examined trait pairs exhibited significant correlations ($\rho_G$ range: -0.57 [AN-q and PFS] to 0.79 [AN-q and WC]), while it is suggestive in nature between AN-q and HDL-C. None of the environmental correlations were significant. Most of these correlations were strongly attenuated when BMI was used as a covariate in the bivariate analyses, but not for HOMA-IR (S1 Fig). In our bivariate genetic analyses (Table 2), the assumption of complete pleiotropy between AN-q and a given CMRF was rejected for all traits (p -value range: $9 \times 10^{-9}$ to 0.001), in turn providing evidence for incomplete pleiotropy.

In regard to the mediation analyses, overall, partial mediation models yielded substantially better indexes of goodness-of-fit compared with full mediation models. To start with, we examined the complex causal relationships among the troika of AN-q, HOMA-IR, and obesity (i.e., BMI) using information from genetic and environmental correlations (i.e., standardized beta coefficients). Of the models tested, as shown in Fig 2 panels A and B, a partial mediation model where BMI explained the relationship with both HOMA-IR and AN-q simultaneously had a better goodness-of-fit than the indirect mediation model that considered BMI with an effect on AN-q through HOMA-IR (BMI→HOMA-IR→AN-q). As can be seen in the partial mediation model, the effect of HOMA-IR was completely attenuated to explain AN-q. Since HOMA-IR showed non-significant effect on AN-q (HOMA-IR→AN-q), we opted not to consider this association further in evaluating other models. The partial mediation model was

**Table 1. Characteristics of the 670 nondiabetic children and adolescents who participated in the SAFARI study, association between Acanthosis Nigricans (AN) and overweight, obese, prediabetes, and metabolic syndrome conditions, and heritability estimate for Acanthosis Nigricans severity scores (AN-q).**

| Variable[a] | N | Mean ± SD or % | Acanthosis nigricans (AN) | | | |
|---|---|---|---|---|---|---|
| | | | % | OR[c] | 95% CI[c] | P value |
| Girls | 670 | 49.3 | - | - | - | - |
| Age (years) | 670 | 11.5 ± 3.5 | - | - | - | - |
| BMI (kg/m$^2$) | 670 | 24.7 ± 6.5 | - | - | - | - |
| Normal weight | 670 | 47.3 | 10.6 (33/278) | 0.11 | 0.07–0.16 | <0.001 |
| Overweight | 670 | 19.1 | 53.1 (186/164) | 9.6 | 6.3–14.5 | <0.001 |
| Obese | 670 | 33.6 | 65.3 (145/77) | 9.3 | 6.4–13.5 | <0.001 |
| Prediabetes | 630 | 13.2 | 47.0 (39/44) | 1.9 | 1.2–3.0 | 0.008 |
| Metabolic syndrome (MS) | 625 | 18.7 | 67.0 (77/38) | 5.7 | 3.7–8.8 | <0.001 |
| Acanthosis nigricans (AN) | 661 | 33.1 | - | - | - | - |
| AN severity score (AN-q)[d] | 660 | 1.3 ± 1.7 | - | - | - | - |

[a]See text for definitions, and the obesity/CMRFs used in the study were described previously by Fowler et al. 2013, and some were adapted from it for description in this table

[b]adjusted for the age, sex, and pubertal-status effects

[c]OR = odds ratio and 95% CI = 95% Confidence Intervals. The ORs were calculated with a total of 661 children with complete data

[d]Heritability ($h^2$) of AN-q = 0.75 (please see text.)

considered as the best model (AIC = 16.8 and BIC = 16.9) compared to the indirect mediation model (AIC = 261.3 and BIC = 261.4) since it was found to fit the data well.

As depicted in Fig 2 panels C and D, using complex models using information from genetic and environmental correlations (Table 2), BMI was associated with AN-q and HOMA-IR mediating most of the associations with CMRFs; but no relationship was found between HOMA-IR and AN-q. The complex associations of the three interrelated variables (HOMA-IR, BMI and AN-q) with other CMRFs (CRP, blood pressure, lipids and PFS) were analyzed by partial and indirect effects of BMI on CMRFs based on their genetic and environmental correlations (i.e., standardized beta coefficients). The models with partial mediation effects, based on genetic or environmental components, exhibited the best goodness-of-fitness indexes compared with the models with mediation effects (Genetic partial mediation vs. Genetic mediation: AIC and BIC = 10,380 vs. 15,105; and, Environmental partial mediation vs. Environmental mediation = 11,904 vs. 15,112). The two models of partial mediation (i.e., genetic [Panel C] and environmental [Panel D]) for BMI effects on metabolic and physical fitness traits are shown in Fig 2. The genetic mediation model (Fig 2 - Panel C) shows a web of BMI partial effects mediated by insulin resistance (HOMA-IR) and AN-q on CMRFs, in turn suggesting the potential activation of other genetic factors influencing inflammation, lipids and physical fitness. When the environmental correlations were analyzed, the model supports BMI as the main factor (with a small mediation effect of HOMA-IR on triglycerides) that influences the cardiometabolic and physical fitness traits.

## 4. Discussion

Acanthosis nigricans (AN) is a strong correlate of obesity and insulin resistance, and those affected with it are at higher risk of developing T2DM. Our data reveals that Mexican American children with AN are significantly, highly likely to be overweight/obese and to have metabolic syndrome or prediabetes. Almost all quantitative traits related to obesity and metabolic syndrome considered for this study are significantly associated with acanthosis nigricans (i.e.,

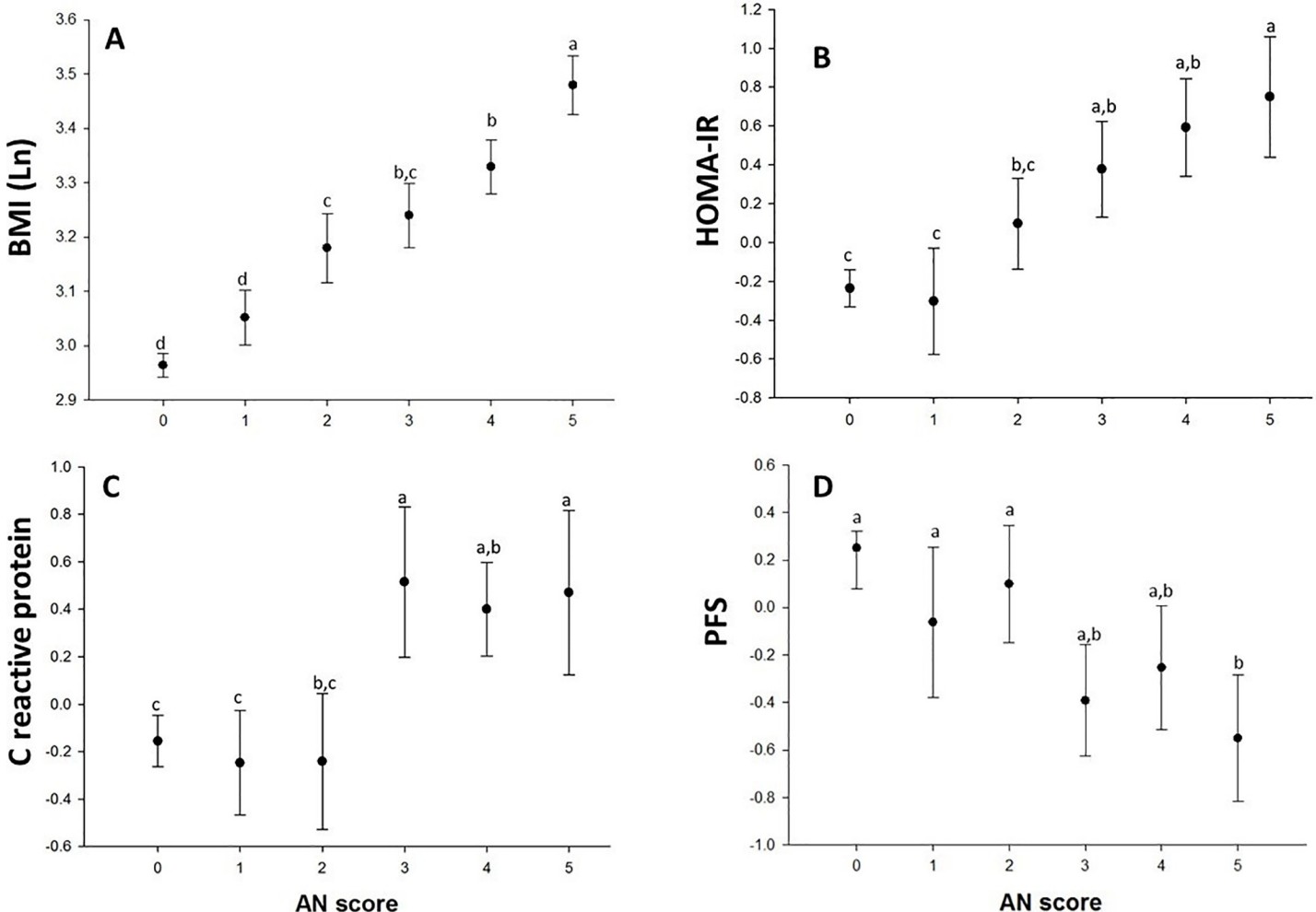

**Fig 1. Panel A.** Positive association between BMI and AN-q. **Panel B.** HOMA-IR showed no difference between AN-q grade 0 vs. AN-q grade 1, but from AN-q grade 2 and so forth showed a positive relationship. **Panel C.** CRP showed no linear trend, but those with AN-q grade 3 and higher had elevated levels compared with lower scores. **Panel D.** PFS was inversely related with AN-q.

AN-q). Similar findings have been reported by other studies. For example, in a study of Latin-American children with obesity, the prevalence rates of AN, IR, metabolic syndrome were found to be 60.6%, 39.4%, and 36.1%, respectively. The prevalence of AN was found to be 70% in the insulin resistant group, while it was 54% in the insulin sensitive group [19]. Our study provides additional support to the statement that AN is highly heritable, and it is strongly associated with obesity-related risk factors, phenotypically and genetically [10]. However, since the direct causal relationship of insulin resistance (IR) with AN in obesity is debatable, this study determined genetic and environmental correlations between AN-q and CMRFs and examined the complex causal relationships among the troika of AN, obesity, and IR using data from children and adolescents from MA families.

To evaluate the potential causal relationships, a causation network approach was employed, which considered BMI effect on AN (i.e., AN-q) and IR as mediators associated with other CMRFs. Mediation analysis has been increasingly used in epidemiological [20] and genetic studies [21]. In our study, for example, BMI through IR was found to have remarkable environmental effects on lipid concentrations and physical fitness, based on a partial mediation

**Table 2. Phenotypic ($\rho_P$), genetic ($\rho_G$), and environmental ($\rho_E$) correlations between Acanthosis Nigricans severity scores (AN-q) and obesity-related traits.**

| Trait | $\rho_P$ | | | $\rho_G$ | | | $\rho_E$ | | |
|---|---|---|---|---|---|---|---|---|---|
| | PE | SE | p-Value | PE | SE | p-Value | PE | SE | p-Value |
| BMI (kg/m$^2$)[a] | 0.65 | 0.025 | $1 \times 10^{-58}$ | 0.72 | 0.07 | $2 \times 10^{-08}$ | 0.36 | 0.27 | 0.321 |
| WC (mm) | 0.66 | 0.020 | $9 \times 10^{-63}$ | 0.79 | 0.07 | $1.1 \times 10^{-07}$ | 0.45 | 0.17 | 0.101 |
| SBP (mm Hg) | 0.20 | 0.041 | $1.8 \times 10^{-06}$ | 0.30 | 0.12 | 0.022 | -0.04 | 0.25 | 0.886 |
| DBP (mm Hg) | 0.14 | 0.040 | 0.0011 | 0.30 | 0.13 | 0.026 | -0.20 | 0.25 | 0.393 |
| TG (mg/dl)[a] | 0.31 | 0.039 | $9.8 \times 10^{-13}$ | 0.41 | 0.11 | 0.0008 | -0.09 | 0.35 | 0.822 |
| HDL-C (mg/dl) | -0.22 | 0.042 | $3.0 \times 10^{-07}$ | -0.25 | 0.13 | 0.064 | -0.17 | 0.25 | 0.528 |
| FG (mg/dl)[a] | 0.02 | 0.043 | 0.6603 | -0.02 | 0.17 | 0.923 | 0.07 | 0.19 | 0.717 |
| FI (μIU/ml)[a] | 0.31 | 0.039 | $2 \times 10^{-13}$ | 0.46 | 0.13 | 0.0013 | 0.06 | 0.22 | 0.802 |
| HOMA-IR[b] | 0.31 | 0.039 | $2 \times 10^{-13}$ | 0.40 | 0.12 | 0.003 | 0.14 | 0.23 | 0.591 |
| CRP (ng/ml)[b] | 0.26 | 0.042 | $3.5 \times 10^{-09}$ | 0.38 | 0.16 | 0.032 | 0.17 | 0.19 | 0.412 |
| PFS[b] | -0.35 | 0.041 | $3.4 \times 10^{-14}$ | -0.57 | 0.13 | 0.0001 | 0.15 | 0.29 | 0.578 |

[a]Traits were log transformed for the genetic analyses

[b]traits were inverse normalized for the genetic analyses. All traits were adjusted for sex, age, and pubertal-status effects; BMI = body mass index; WC = waist circumference; SBP = systolic blood pressure; DBP = diastolic blood pressure; TG = triglycerides; HDL-C = high density lipoprotein cholesterol; FG = fasting plasma glucose; FI = fasting insulin; HOMA-IR = homeostasis model of assessment-insulin resistance; CRP = high-sensitivity C-reactive protein; PFS = Harvard physical fitness score.

model. The partial mediation model based on information from genetic effects exhibited a complex network of relationships between BMI, IR, AN, and other CMRFs. Interestingly, in contrast to some previous reports associated with malignancy and insulin receptor abnormalities, as described below, our findings failed to support the notion that IR causally explains benign AN; the partial mediation model where BMI explained both IR and AN had the best goodness of fit. This lack of direct effect between IR and AN is consistent with previous reports [22,23]. The mediation analysis also confirms that obesity (i.e., BMI) had partial effect, not only on IR and AN, but also on inflammation, blood pressure, lipids and physical fitness. Our findings are in line with a recent report that found the association of AN with hyperinsulinemia was attenuated after accounting for BMI influences in adolescents [11].

AN was initially observed association in malignancy [24] and later, Kahn and colleagues described extreme IR due to insulin receptor abnormalities (Type A IR) [25]. This excess insulin activates the insulin-like growth factor-1 receptors or fibroblast growth factor receptors, which has been implicated in the pathogenesis of AN [26]. 30 Interestingly, the initial classification of AN in subjects with obesity was defined as pseudoacanthosis, because its etiology and clinical manifestations were different from AN caused by insulin receptor mutations. However, subsequently, AN definition has been modified to include obesity, and the usage of pseudoacanthosis was discarded. Given the pandemic of obesity in global populations, the prevalence of AN has increased. In some recent studies, it was found to be quite high, ranging from 5% to 74% [2,27]. 31 Our study reiterates the notion that AN is strong correlate of obesity, and reveals the direct effect of obesity (i.e., BMI) than an indirect mediation through IR (i.e., HOMA-IR) on AN (i.e., AN-q). It is possible that the skin photo phenotype may have a modificatory effect on the IR and AN association [5]. Berstein et al. [28] found a lack of increase of pituitary peptides in obesity associated with AN, but this study included small sample size and the authors suggested further evaluations.

Insulin resistance and AN are partial mediators of BMI with genetic and environmental sharing influences on CRP, blood pressure, lipids, and physical fitness. The severity of AN is directly proportional to obesity, hyperinsulinemia, and metabolic syndrome-related risk

A. Indirect Mediation Model

B. Partial Mediation Model
(Best goodness-of-fit)

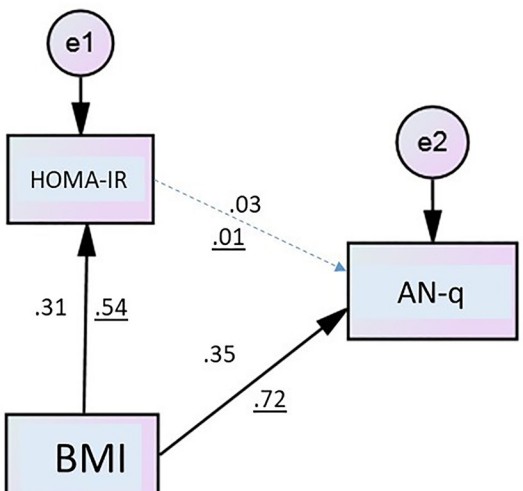

C. Genetic Partial Mediation

D. Environmental Partial Mediation

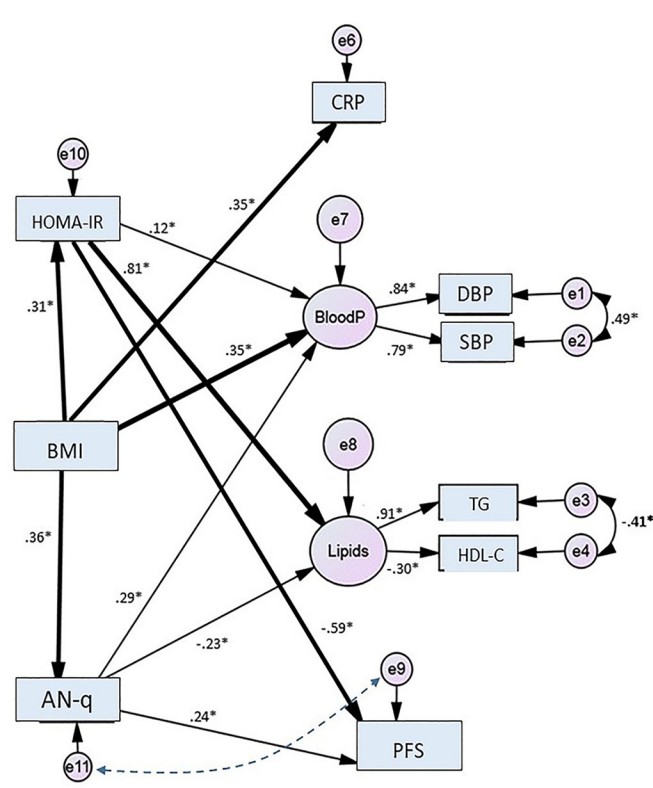

**Fig 2. Panel A**. Indirect mediation model shows BMI had an effect on AN-q through HOMA-IR. **Panel B**. Partial mediation model, the effect of HOMA-IR was completely attenuated to explain AN-q. This model fits the data well (The numbers correspond to environmental correlations and underlined numbers correspond to genetic correlations). Panel C and D are two models of partial mediation for BMI effects on metabolic and PFS traits. The thick arrows show standardized beta-coefficients greater than 0.3; thin lines show those beta coefficients between 0.3 and 0.1. Coefficients lower than 0.1 are not shown. The boxes indicate observed variables, circles indicate latent endogenous variables, those with a letter "e" are error terms. **Panel C** (genetic partial mediation) shows a web of genetic relationships. HOMA-IR and AN-q are mediators of BMI, perhaps activating other genes to have an effect on inflammation, lipids and physical fitness. **Panel D** (environmental

partial mediation) supports the hypothesis that BMI is the main variable (with a small mediation effect of HOMA-IR on triglyceride concentrations) that influences the metabolic and PFS traits. This model had the best goodness-of-fit tests compared with other models of indirect effect of BMI by HOMA-IR or AN-q as mediators; * p-value < 0.01. BloodP: Blood pressure (latent factor). The dotted lines between AN-q and PFS show bidirectional (correlation) association between these two variables in an independent model.

factors which is also in line with the previous studies [2,8,29–32]. Chronic inflammation associated with obesity explains the predominant effect of obesity relative to the other metabolic measures. In contrast, the IR as assessed by HOMA-IR, has minor influence on CRP concentrations in SAFARI children. The strong genetic correlation of AN with obesity measures (body composition and abdominal obesity) [33], was also paralleled by measures of insulin resistance and sensitivity [19] indicating that AN, obesity, and IR are strongly influenced by common genetic or pleiotropic influences in our study sample of Mexican American children. The pattern of correlations involving IR are similar to those reported in previous studies [10,31–33]. In regard to CRP (a marker of inflammation), it is interesting to note that the transition between AN grade 2 and 3 appears to be a threshold for chronic inflammation. It is worthwhile to note that lypodistrophies, a group of diverse rare disorders, are associated with various disease conditions such as dyslipidemia, insulin resistance [34,35] and often with skeletal muscle abnormalities, like myoedema [36]. 40 It is also associated with AN via insulin resistance. The lypodistrophies have been described in Mexicans [37].

Aside from the debate that whether IR is a direct contributor to AN, it is evident from this study that AN is associated with a number of cardiometabolic risk factors through complex genetic and environmental relationships. Thus, AN is shown to be a compound marker of metabolic risk in children and adolescents. AN finding provides a simple non-invasive clinical tool that can be used for pediatric screening of cardiometabolic risks in the public health initiatives. It is currently implemented as the Texas Risk Assessment for Type 2 Diabetes in Children (TRAT2DC) program, a legislatively mandated program developed, coordinated, and administered by the University of Texas Rio Grande Valley Border Health Office. It assesses students across the schools in Texas to identify children at high risk of developing T2DM and alerts the parents of children who are found to be at risk of T2DM and other health conditions. During vision/hearing and scoliosis screenings of 1st, 3rd, 5th and 7th graders in public and private schools, certified individuals assess children for AN. Children who are identified with AN undergo additional assessments of BMI, BMI percentile, and blood pressure. As examples, here below we provide information from three independent school districts (ISDs) in the Rio Grande Valley area, where more than 90% of the population is Hispanic. During the 2017–2018 period, the prevalence of AN in Donna ISD was 13.9% (obesity = 80.0% and hypertension = 32.6%), and 5.7% (obesity = 93.7% and hypertension = 35.5%) in Mission ISD. The prevalence in Pharr-San Juan-Alamo ISD was 15.8% (obesity = 83.8% and hypertension = 32.0%).

In conclusion, obesity is shown to be a direct contributor to AN, hyperinsulinemia, and other CMRFs in Mexican American children. Moreover, AN is found to be a compound marker of complex cardiometabolic risk, which can be valuable, simple non-invasive clinical assessment tool for public health screening to assess cardiometabolic risk in children as exemplified by our TRAT2DC program. Early identification of AN in children should help to prevent or delay future health problems for children at risk of T2DM and its associated health conditions. Also, the findings form our mediation analyses mirror a complex interwork of relationships between obesity-related traits including AN, and IR and AN as partial mediators of BMI. There appears to be an immediate need for identifying the molecular mechanisms and

pathways underlying these complex relationships among the Obesity/AN related phenotype examined in this study.

## Supporting information

**S1 Table. Detailed number of pairs classified by relationship.**
(DOCX)

**S1 Fig. Functions of selected metabolic traits by acanthosis nigricans score (ANc).** Panels A to D represent systolic blood pressure; E to H hsCRP; I to L triglycerides, and M to P PFS (Harvard test). The phenotypic function by ANc adjusted by sex, age and family is shown in the panels A, E, I and M. These functions shows clear positive or inverse relationship with ANc. The attenuation by BMI is shown in panels C, G, K and O; the hsCRP was maintained in a cubic polynomial and triglycerides on squared. The hsCRP shows a critical point between scores 2 to 3 showing a clear increase on the concentration; meanwhile triglycerides was positive for both BMI and ISI variables. When analyzed with the two covariates, all variables were attenuated.
(TIF)

**S1 Data. RGen_corr_matrix.csv // Matrix of genetic correlations between traits.**
(CSV)

**S2 Data. REnv_corr_matrix.csv // Matrix of environmental correlations between traits.**
(CSV)

**S3 Data. Residuals.csv // Residuals after polygenic regression adjusted by sex, age and their interactions.**
(CSV)

## Acknowledgments

JCLA, GC, SFDP, and RD designed the research, conducted the research, analyzed the data, wrote the manuscript, and helped with the manuscript going through the peer-review process. SP, VSF, SPF, RA, and CPJ conducted the research, contributed to data analysis and provided critical review of the manuscript. RGR, ADB and JHR conducted the research and contributed to the manuscript preparation. DS, DDG, SM and JLL conducted research and reviewed/ edited the manuscript. RAD, DML, JB, DEH and RD designed the research, provided study oversight, contributed to data analysis, and reviewed/edited the manuscript. All authors critically read, revised, and approved the final version of the manuscript before its submission. We thank Dr. William Rogers, Dr. Rolando Lozano, Richard Granato, Margaret Fragoso, David Rupert, Rhonda Lyons, Tanya Prado, Elizabeth Sosa, Bonnie Sanchez, Ram Prasad Upadhayay, and Nicolas Ballí for their excellent help and assistance. We thank the University Health System and the Texas Diabetes Institute for extending their clinical research facilities to the SAFARI study. Lastly and most importantly, we are deeply indebted to the children, teenagers, parents, and extended family members of the SAFARI study, whose great enthusiasm and commitment have made this research possible.

## Author Contributions

**Conceptualization:** Juan C. Lopez-Alvarenga, Geetha Chittoor, Solomon F. D. Paul, Ralph A. DeFronzo, John Blangero, Daniel E. Hale, Ravindranath Duggirala.

**Data curation:** Roy G. Resendez, Joselin Hernandez-Ruiz, Alvaro Diaz-Badillo.

**Formal analysis:** Juan C. Lopez-Alvarenga, Geetha Chittoor, Sobha Puppala, Vidya S. Farook, Sharon P. Fowler, Rector Arya, Ravindranath Duggirala.

**Funding acquisition:** Ravindranath Duggirala.

**Investigation:** Roy G. Resendez, Joselin Hernandez-Ruiz.

**Methodology:** Solomon F. D. Paul, Joselin Hernandez-Ruiz, Alvaro Diaz-Badillo, David Salazar, Doreen D. Garza, Donna M. Lehman, Srinivas Mummidi, Rector Arya, Christopher P. Jenkinson, Ravindranath Duggirala.

**Project administration:** Roy G. Resendez, Ravindranath Duggirala.

**Resources:** Jane L. Lynch, Ravindranath Duggirala.

**Supervision:** Ralph A. DeFronzo, John Blangero, Daniel E. Hale, Ravindranath Duggirala.

**Validation:** Juan C. Lopez-Alvarenga, Sharon P. Fowler, Alvaro Diaz-Badillo, David Salazar, Doreen D. Garza, Ravindranath Duggirala.

**Visualization:** Juan C. Lopez-Alvarenga.

**Writing – original draft:** Juan C. Lopez-Alvarenga, Geetha Chittoor, Solomon F. D. Paul, Srinivas Mummidi, Ravindranath Duggirala.

**Writing – review & editing:** Sobha Puppala, Vidya S. Farook, Sharon P. Fowler, David Salazar, Doreen D. Garza, Donna M. Lehman, Rector Arya, Christopher P. Jenkinson, Jane L. Lynch, Ralph A. DeFronzo, John Blangero, Daniel E. Hale.

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
