## [Decision Letter · Decision Letter 0]

8 Jun 2020

PONE-D-20-02036

Acanthosis Nigricans as a Composite Marker of Cardiometabolic Risk and Its Complex Association with Obesity and Insulin Resistance in Mexican American Children

PLOS ONE

Dear Dr. Juan Carlos Lopez-Alvarenga,

Thank you for submitting your manuscript to PLOS ONE. After careful consideration, we feel that it has merit but does not fully meet PLOS ONE’s publication criteria as it currently stands. Therefore, we invite you to submit a revised version of the manuscript that addresses the points raised during the review process.

The reviewer(s) have suggested some revisions to your manuscript. Therefore, I invite you to respond to the reviewer(s)' comments and revise your manuscript.

We look forward to receiving your revised manuscript.

Kind regards,

Fernando Guerrero-Romero, MD, PhD

Academic Editor

PLOS ONE

Journal Requirements:

'..This study was supported by grants from the National Institutes of Health (R01 HD049051, HD041111, DK053889, DK042273, K01DK064867, P01 HL045522, DK047482, MH059490, M01-RR-01346, and HD049051-5S1 [ARRA]). This work was also supported by a Veterans Administration Epidemiologic grant to R.A.D...  The AT&T Genomics Computing Center supercomputing facilities used for this work were supported in part by a gift from the AT&T Foundation and with support from the National Center for Research Resources Grant Number S10 RR029392. This investigation was conducted in facilities constructed with support from Research Facilities Improvement Program grants C06 RR013556 and C06 RR017515 from the National Center for Research Resources of the National Institutes of Health...'

'RD received grants: R01 HD049051, HD041111, DK053889, DK042273, K01DK064867, P01 HL045522, DK047482, MH059490, M01-RR-01346, and HD049051-5S1 [ARRA]'

Please clarify the sources of funding (financial or material support) for your study. List the grants or organizations that supported your study, including funding received from your institution.State what role the funders took in the study. If the funders had no role in your study, please state: “The funders had no role in study design, data collection and analysis, decision to publish, or preparation of the manuscript.”If any authors received a salary from any of your funders, please state which authors and which funders.

Reviewers' comments:

Reviewer's Responses to Questions

**Comments to the Author**

1. Is the manuscript technically sound, and do the data support the conclusions?

Reviewer #1: Partly

Reviewer #2: Yes

2. Has the statistical analysis been performed appropriately and rigorously? 

Reviewer #1: Yes

Reviewer #2: Yes

3. Have the authors made all data underlying the findings in their manuscript fully available?

Reviewer #1: Yes

Reviewer #2: Yes

4. Is the manuscript presented in an intelligible fashion and written in standard English?

Reviewer #1: Yes

Reviewer #2: Yes

5. Review Comments to the Author

Reviewer #1: This manuscript describes the relationship between obesity, insulin resistance (IR), and acanthosis nigricans (AN). The relationship among these factors is not novel, but the authors have a novel statistical approach using many variables, genetics, and phenotypic. They found that obesity explains the association of IR with AN, but no causal relationship between IR and AN in Mexican American children. The results of this study could be important in clinical practice due to the AN severity-classification, which is low-cost, and it seems to be easy to measure, is correlated with cardiometabolic factors. However, the manuscript has critical methodological issues that need clarification and to be revised.

Main Concerns

1. The kinship of the sample is not describing in the manuscript; I mean the pairs of full or half-brothers, cousins, etc. It is not clear which kind of parentage was used for hereditability estimation; in other words, it is not clear which kinship measures fed the model. It is also not clear if hereditability is the only genetic variable used in the study.

2. Describe criteria cut-off for prediabetes and cardiometabolic factors; cut-of for dichotomized IFG, IGT, HDL-C, blood pressure and son on.

3. Which were the environmental variables used? Those need clarification.

4. The authors mention three methods to choose the best model (goodness-of-fit statistics, Akaike information criterion, and Bayes information). Still, they did not specify the threshold criterion to select the models. For example, mention the difference in the AIC values. This is a concerning matter for the reproducibility and the credibility of the chosen best model.

5. Mediation analysis has enough statistical power, but what about with the variance components analysis?

6. For clinical practice, are the pediatricians familiarized with the AN severity scale used in this study? If they are not, extending the AN as a proxy of cardiometabolic factors could be a limitation.

Reviewer #2: Please add a brief discussion on the ethnicity of the experimental group, in addition to describing them as Mexican Americans please provide more information on the ethnic ancestry of their two previous generations. It is also relevant to know if they were born in the US, if they are US citizens, or if they were born in Mexico and migrated to the US. This would provide a better understanding on the possible applications of the reported results to Mexican children living in urban areas similar to the state of Texas, such as the nearby city of Monterrey, Nuevo Leon, Mexico.

6. PLOS authors have the option to publish the peer review history of their article (what does this mean?). If published, this will include your full peer review and any attached files.

Reviewer #1: No

Reviewer #2: Yes: Julio Granados

---

## [Editor Report · Decision Letter 1]

28 Sep 2020

Acanthosis Nigricans as a Composite Marker of Cardiometabolic Risk and Its Complex Association with Obesity and Insulin Resistance in Mexican American Children

PONE-D-20-02036R1

Dear Dr. Lopez-Alvarenga,

We’re pleased to inform you that your manuscript has been judged scientifically suitable for publication and will be formally accepted for publication once it meets all outstanding technical requirements.

Kind regards,

Fernando Guerrero-Romero, MD, PhD

Academic Editor

PLOS ONE
---

## [Editor Report · Acceptance letter]

2 Oct 2020

PONE-D-20-02036R1 

Acanthosis nigricans as a composite marker of cardiometabolic risk and its complex
association with obesity and insulin resistance in Mexican American children 

Dear Dr. Lopez-Alvarenga:

I'm pleased to inform you that your manuscript has been deemed suitable for publication in PLOS ONE. Congratulations! Your manuscript is now with our production department. 

Kind regards, 

on behalf of

Dr Fernando Guerrero-Romero 

Academic Editor

PLOS ONE